# Nickel-catalyzed electrophiles-controlled enantioselective reductive arylative cyclization and enantiospecific reductive alkylative cyclization of 1,6-enynes

Wenfeng Liu[1,3], Yunxin Xing[2,3], Denghong Yan[2,3], Wangqing Kong [1] ✉ & Kun Shen [2] ✉

Transition metal-catalyzed asymmetric cyclization of 1,6-enynes is a powerful tool for the construction of chiral nitrogen-containing heterocycles. Despite notable achievements, these transformations have been largely limited to the use of aryl or alkenyl metal reagents, and stereoselective or stereospecific alkylative cyclization of 1,6-enynes remains unexploited. Herein, we report Ni-catalyzed enantioselective reductive *anti*-arylative cyclization of 1,6-enynes with aryl iodides, providing enantioenriched six-membered carbo- and heterocycles in good yields with excellent enantioselectivities. Additionally, we have realized Ni-catalyzed enantiospecific reductive *cis*-alkylative cyclization of 1,6-enynes with alkyl bromides, furnishing chiral five-membered heterocycles with high regioselectivity and stereochemical fidelity. Mechanistic studies reveal that the arylative cyclization of 1,6-enynes is initiated by the oxidative addition of Ni(0) to aryl halides and the alkylative cyclization is triggered by the oxidative addition of Ni(0) to allylic acetates. The utility of this strategy is further demonstrated in the enantioselective synthesis of the antiepileptic drug Brivaracetam.

Optically pure nitrogen-containing heterocycles have long been of great interest due to their frequent occurrence in natural products with diverse pharmaceutical activities[1]. For example, chiral 2-pyrrolidones and 1,2,3,6-tetrahydropyridines constitute core structural elements in medicinally relevant compounds. In particular, Brivaracetam and Seletracetam are used for the treatment of partial-onset seizures (Fig. 1a)[2,3]. The ergot alkaloids lysergic acid and lysergol exhibit a wide range of biological activities, and their synthetic derivatives, such as bromocriptine and pergolide, are used as antiprolactin and anti-Parkinson's disease drugs[4,5]. As a consequence, the development of efficient and enantioselective methods to access these important chiral nitrogen-containing heterocyclic scaffolds is highly desirable and sought after.

Transition metal-catalyzed asymmetric cyclization of 1,6-enynes involves carbometallation, cyclization, and β-elimination and is a powerful tool for the construction of chiral nitrogen-containing heterocycles[6–11]. The Lu group[12–14] and the Murakami group[15,16] pioneered the study on the Pd- and Rh-catalyzed asymmetric *cis*-arylative cyclization of 1,6-enynes with arylboronic acids to form five-membered heterocycles, respectively. Very recently, our group developed a Ni-catalyzed asymmetric arylative cyclization of fluoroalkyl-substituted 1,6-enynes with arylboronic acids for the synthesis of Seletracetam[17].

[1]The Institute for Advanced Studies (IAS), Wuhan University, Wuhan 430072, China. [2]Department of Radiology, Zhongnan Hospital of Wuhan University, School of Pharmaceutical Sciences, Wuhan University, Wuhan 430071, China. [3]These authors contributed equally: Wenfeng Liu, Yunxin Xing, Denghong Yan. ✉e-mail: wqkong@whu.edu.cn; kun.shen@whu.edu.cn

**Fig. 1 | Transition metal-catalyzed functionalization/cyclization of 1,6-enynes. a** Bioactive molecules containing chiral 2-pyrrolidone and 1,2,3,6-tetrahydropyridine skeletons. **b** Enantioselective cyclization of 1,6-enynes via carbometallation/cyclization/elimination pathway.

Alternatively, the Lam group described an elegant Ni-catalyzed asymmetric *anti*-arylative cyclization of 1,6-enynes with arylboronic acids to afford six-membered heterocycles[18] via reversible alkenylnickel *E/Z* isomerization[19–25]. These redox-neutral reactions were proposed to be initiated by carbometallation of the alkyne moiety, followed by cyclization and β-heteroatom elimination. Although remarkable achievements have been made in this field, there are still some considerable limitations. (1) These transformations are essentially restricted to the use of aryl or alkenyl metal reagents with limited functional group compatibility. (2) Stereoselective or stereospecific alkylative cyclization of 1,6-enynes remains unexploited, probably due to the easy β-H elimination of the alkyl metal intermediate (Fig. 1b).

On the other hand, transition metal-catalyzed asymmetric allylic substitution reactions are an important method for C–C bond formation and have been widely used in the synthesis of natural products[26–30]. In this context, the enantiospecific allylic functionalization takes advantage of the ready availability of highly enantioenriched allylic alcohols while avoiding the need for chiral ligands, providing facile access to a range of enantioenriched products with alkene functional handles for down-stream elaboration[31,32]. Such transformation involves the oxidative addition of a low-valent metal (Pd[33–37], Cu[38–42], Rh[43], and Ni[44–46]) to allylic electrophiles to generate π-allylic intermediates, followed by coupling with various metal reagents such as organomagnesium, organozinc, organoaluminium, and organoboron reagents (Fig. 2a).

Over the past decade, there has been a surge of interest in Ni-catalyzed reductive cross-coupling reactions, a strategy that allows reactions to be performed under mild conditions with high functional group tolerance and avoids the handling of sensitive organometallic reagents (Fig. 2b)[47–57]. This strategy has emerged as an efficient and practical method for the enantioselective coupling of alkyl electrophiles[58–62]. Moreover, the Jarvo group developed the enantiospecific intramolecular reductive coupling of alkyl halides for the preparation of enantioenriched cyclic compounds[63,64].

Inspired by the attractive Ni-catalyzed reductive cross-coupling approach, we envisaged that the reaction of alkyne-tethered allylic acetate **1** with alkyl bromide is initiated by the formation of a π-allylnickel intermediate, followed by intramolecular insertion into the alkyne and cross-coupling with the alkyl halide, which would lead to the formal alkylative cyclization product (Fig. 2c). Successful implementation of this strategy would address the challenging problem of stereoselective alkylative cyclization of 1,6-enynes. Although non-asymmetric reductive allylation reactions have been extensively studied[65–72] and the Co-catalyzed enantiospecific reductive vinylation of allylic alcohols with vinyl triflates was developed by Shu[73], to the best of our knowledge, stereoselective reductive allylic alkylation remains unexplored.

To achieve the conceptually simple yet attractive transformation described above, many challenging issues need to be addressed. (1) The oxidative addition of allylic electrophiles to low-valent nickel is highly competitive with that of aryl halides, thus leading to a chemoselective problem. (2) Direct coupling between organohalides and allylic electrophiles is also an important complicating factor. (3) As reported by Lam[18,19] and Montgomery[74], alkenylnickel species are prone to *E/Z* isomerization, which may result in a mixture of *cis/trans* isomers. (4) Another challenging issue is to control the enantioselectivity or enantiospecificity of the cyclization process.

Herein, we report our recent findings on the asymmetric reductive cyclization of 1,6-enynes with organohalides (Fig. 2c). The enantioselective reductive *anti*-arylative cyclization of 1,6-enynes with aryl iodides was achieved, providing enantioenriched six-membered carbo- and heterocycles in good yields with excellent enantioselectivities. Moreover, the enantiospecific reductive *cis*-alkylative cyclization of 1,6-enynes with alkyl bromides has also been realized, furnishing chiral tetrahydropyrroles in good yields with high regioselectivity and stereochemical fidelity. Mechanistic studies reveal that the arylative cyclization of 1,6-enynes is initiated by the oxidative addition of Ni(0) to aryl halide, followed by carbonickelation of the alkyne, *E/Z* isomerization and enantioselective addition to allylic acetates. The alkylative cyclization is triggered by the oxidative addition of Ni(0) to allylic acetates to generate π-allylic intermediates, followed by stereospecific insertion of alkynes and coupling with alkyl halides. The utility of this strategy was further demonstrated in the enantioselective synthesis of the antiepileptic drug Brivaracetam.

**Fig. 2 | Asymmetric reductive cyclization of 1,6-enynes. a** Enantiospecific allylic substitution with organometallic reagents. **b** Enantioselective or enantiospecific reductive cross-coupling of alkyl halides. **c** Enantioselective or enantiospecific reductive cyclization of 1,6-enynes (this work).

## Results

### Reaction development

We initiated our investigation by exploring the enantioselective reductive arylative cyclization of alkyne-tethered (*Z*)-allylic acetate **1a** with PhI **2a** (Table 1). After extensive investigation of all of the reaction parameters, we found that the use of Ni(OTf)$_2$/(*S*)-$^i$Pr-NeoPhox (**L8**) as the precatalyst and Mn as the reducing agent in DMA/HFIP at 80 °C afforded the desired tetrahydropyridine **3aa** in 74% yield with 95% ee (entry 1). Changing (*S*)-$^i$Pr-NeoPhox (**L8**) to other chiral P,N-ligands such as (*S*)-R-Phox ligands (**L1**–**L4**), (*S,R*)-In-Phox (**L5**), (*S,S$_p$*)-$^i$Pr-Phosferrox (**L6**), and (*S*)-$^t$Bu-NeoPhox (**L7**) resulted in either lower yields or lower enantioselectivities (entries 2-8). Using other nickel catalysts such as NiBr$_2$•glyme and Ni(OAc)$_2$•4H$_2$O or reducing the catalyst loading resulted in lower yields, but the enantioselectivity of **3aa** remained unchanged (entries 9-11). Using Zn instead of Mn gave **3aa** in 49% yield with 95% ee (entry 12). Notably, the co-solvent HFIP played a crucial role in achieving high yield and enantioselectivity. Both the yield and enantiomeric excess of **3aa** were dramatically decreased in the absence of HFIP (entry 13). Other protonic co-solvents, such as TFE and $^i$PrOH, gave diminished yields and enantioselectivities (entries 14–15).

With optimal reaction conditions in hand, we turned our attention to examining the scope of enantioselective reductive arylative cyclization of 1,6-enynes (Fig. 3). A wide range of diversely substituted aryl iodides could participate well in the desired reactions to give tetrahydropyridines **3ab**–**3am** in moderate to good yields (48–80%) with excellent enantioselectivities (92–98%). Aryl iodides bearing various synthetic useful functional groups such as methoxy (**3ac**), cyano (**3ad**),

ester (**3ae**), trifluoromethyl (**3af**), fluoride (**3ag**), boronic ester (**3ah**), chloride (**3aj**), and methylenedioxy (**3al**) at the aromatic ring were well tolerated. The substitution pattern of aryl iodides was also investigated. Aryl iodides bearing a methyl group at the *para*- (**3ab**) or *meta*- (**3ai** and **3ak**) position of the aromatic ring underwent the reaction smoothly to provide the corresponding products in 59–80% yields with 93–95% ee. The absolute configuration of **3aj** was unambiguously established by X-ray crystallography, and those of all other *N*-heterocyclic products were assigned accordingly. However, *ortho*-methyl substituted aryl iodide failed to give the desired product, probably due to steric hindrance.

The substrate scope of 1,6-enynes was then investigated. Remarkably, the stereochemistry of allylic acetate has little effect on the reaction outcome. As shown, the (*E*)-1,6-enyne (*E*)-**1a** reacted smoothly with PhI to provide the expected product **3aa** in 83% yield and 87% ee. It is worth mentioning that the *Z*-stereochemistry of the alkene moiety is crucial for this addition-cyclization-elimination reaction, whereas *E*-alkenes cannot form the corresponding cyclization products in previously reported redox-neutral strategies[18]. Regarding the substitute on the alkyne moiety, the reaction is compatible with various aromatic groups and provided the corresponding products **3ba**–**3ka** in moderate to high yields (32–77%) with excellent enantioselectivities (90–96% ee). In general, electron-rich arene groups such as 4-methylphenyl (**3ba**), 4-methoxyphenyl (**3ca**), 3-methylphenyl (**3ha**), and 3,4-methxylenedioxyphenyl (**3ja**) gave higher yields than electron-deficient arene groups such as 4-methoxycarbonylphenyl (**3da**), 4-trifluoromethylphenyl (**3ea**), and 4-fluorophenyl (**3fa**). Notably, the reaction tolerates boronate (**3ga**)

**Table 1 | Condition optimization for enantioselective reductive arylative cyclization of 1,6-enynes[a]**

| Entry | Variation from standard conditions | Yield (%) of 3aa[b] | ee (%) of 3aa[c] |
|---|---|---|---|
| 1 | None | 74 | 95 |
| 2 | **L1** instead of **L8** | 44 | 97 |
| 3 | **L2** instead of **L8** | 62 | 90 |
| 4 | **L3** instead of **L8** | 50 | 88 |
| 5 | **L4** instead of **L8** | 42 | 37 |
| 6 | **L5** instead of **L8** | 78 | 91 |
| 7 | **L6** instead of **L8** | 36 | 95 |
| 8 | **L7** instead of **L8** | 28 | 96 |
| 9 | NiBr$_2$·glyme instead of Ni(OTf)$_2$ | 65 | 95 |
| 10 | Ni(OAc)$_2$·4H$_2$O instead of Ni(OTf)$_2$ | 41 | 95 |
| 11 | 10 mol% Ni(OTf)$_2$, 20 mol% (S)-**L8** | 52 | 95 |
| 12 | Zn instead of Mn | 49 | 95 |
| 13 | without HFIP | 53 | 62 |
| 14 | TFE instead of HFIP | 51 | 90 |
| 15 | $^i$PrOH instead of HFIP | 49 | 88 |

[a]Reaction conditions: **1a** (0.2 mmol), **2a** (0.3 mmol), Ni(OTf)$_2$ (15 mol %), (S)-**L8** (30 mol %), and Mn (0.6 mmol) in DMA/HFIP (2 mL/2 mL) in a sealed tube at 80 °C for 12 hours.
[b]Isolated yield after flash chromatography.
[c]The ee values were determined by HPLC on a chiral stationary phase.

and chlorine (**3ia**) groups, offering opportunities for further transformations via cross-coupling manipulations. Replacement of the substituent on the nitrogen from *p*-toluenesulfonyl to *tert*-butoxycarbonyl was also feasible, affording **3la** in 41% yield and 95% ee. Moreover, 1,6-enynes with various backbones were tested. Changing the linking group between the alkyne and the allylic acetate to an oxygen or an all-carbon tether enabled the formation of tetrahydropyran **3ma** and carbocycle **3na** in 51% yield with 94% ee and 65% yield with 92% ee, respectively.

We further expect to realize the nickel-catalyzed reductive alkylative cyclization of 1,6-enynes with alkyl halides. We explored the reductive alkylative cyclization of **1a** with alkyl bromide **5a**. Instead of obtaining the tetrahydropyridine product, we isolated the tetrahydropyrrole product **6aa** in 79% yield as a single regioisomer with excellent stereoselectivity (*E/Z* > 20:1) using the combination of NiBr$_2$ and 2,2'-bipyridine (BPy, **L9**) as the catalyst, and Mn as the reductant (see Supplementary Information Section 4). This result is quite different from Montgomery's report on nickel-catalyzed oxidative cyclization and reductive cross-electrophile coupling of aldehydes, alkynes, and alkyl halide to produce a mixture of *cis-trans* isomers[74]. Encouraged by this result, we screened a series of chiral ligands in an attempt to render the alkylative cyclization reaction asymmetric. However, all our attempts resulted in only racemic products. We speculate that the

reaction mechanism of the alkylative cyclization is different from that of the arylative cyclization and that the former may be initiated by the oxidative addition of allylic acetates to nickel to form π-allylnickel intermediates. We, therefore, turned our attention to the development of nickel-catalyzed enantiospecific alkylative cyclization of 1,6-enynes with alkyl halides.

We started our investigation by exploring the cyclization reaction of alkyne-tethered (*E*)-allylic acetate **4a** (99% ee) with **5a** (Table 2). The reaction was conducted in DMA at 20 °C using 10 mol% of Ni(ClO$_4$)$_2$·6H$_2$O and 20 mol% of BPy (**L9**) as catalyst (entry 1). The tetrahydropyrrole product **7aa** was obtained in 56% yield with a significant decrease in enantiospecificity (5% ee), reinforcing the notion that the enantiospecific alkylative cyclization would be far from trivial. We found that the electronegativity of the ligand has a strong influence on the stereospecificity. Using the pyridine oxazole ligand **L10**, **7aa** could be obtained in 60% yield with 80% ee (entry 2). Surprisingly, both lowering the reaction temperature to 0 °C and increasing the temperature to 40 °C resulted in a significant decrease in the efficiency and stereospecificity of the reaction (entries 3–4). The effect of different solvents was also investigated, and the stereospecificity of **7aa** increased to 86% when THF was used as a co-solvent (entries 5–8). Subsequently, we investigated the effect of electronic and steric modifications of Pybox ligands on reactivity and selectivity (entries

**Fig. 3 | Reaction scope of asymmetric reductive arylative cyclization.** The reactions were performed on a 0.2 mmol scale under the conditions in Table 1, entry 1.

9–14). Adjusting the electronic properties of the pyridine ring did not lead to further improvements (**L10–L12**). A clear trend was observed where increased steric bulk at the 5-position of the oxazoline ring enhanced stereospecificity (**L13**). Adding steric repulsion on the pyridine ring (**L14**) led to very low yield. To our delight, the spirocyclic ligand **L15** was found to be the most effective in terms of both reactivity and stereospecificity. Other nitrogen ligands, including phenanthroline (**L17**) and terpyridine (**L18**), have been widely used in Ni-catalyzed reductive cross-coupling reactions[47–57]; however, they produced trace amounts of the desired product **7aa** and resulted in a significant erosion of the ee values (entries 15–16). The addition of NaI proved to be crucial for the high efficiency of the reaction (entry 17).

Control experiments showed that the reaction did not occur in the absence of Mn[0] and ligand (entries 18–19).

With optimal reaction conditions in hand, we examined the scope of enantiospecific reductive alkylative cyclization of 1,6-enynes with alkyl bromides (Fig. 4). We were a pleasure to find that a variety of alkyl bromides could undergo alkylative cyclization to furnish the tetrahydropyrroles **7aa**–**7ao** in good yields with excellent enantiospecificity. Alkyl bromides with different substituents and functional groups, such as long-chain alkyls (**7aa** and **7ab**), chloride (**7ac**), fluoride (**7ad**), silyl ether (**7ae**), acetoxyl (**7af**), ester (**7ag**), cyanide (**7ah**), boronic ester (**7ai**), acetals (**7aj**), amides (**7ak**–**7am**), alkenyl (**7an**), and alkynyl (**7ao**), were well compatible with the current reaction conditions. The

**Table 2 | Condition optimization for enantiospecific reductive alkylative cyclization of 1,6-enynes[a]**

| Entry | Ligand | Solvent | Yield (%) of 7aa[b] | ee/es (%) of 7aa[c] |
|---|---|---|---|---|
| 1 | L9 | DMA | 56 | 5/5 |
| 2 | L10 | DMA | 60 | 80/81 |
| 3[d] | L10 | DMA | 38 | 52/53 |
| 4[e] | L10 | DMA | 48 | 67/68 |
| 5 | L10 | THF | trace | – |
| 6 | L10 | DMF | 49 | 75/76 |
| 7 | L10 | DMSO | trace | – |
| 8 | L10 | DMA/THF(1/2) | 78 | 86/87 |
| 9 | L11 | DMA/THF(1/2) | 34 | 46/46 |
| 10 | L12 | DMA/THF(1/2) | 77 | 84/85 |
| 11 | L13 | DMA/THF(1/2) | 60 | 29/29 |
| 12 | L14 | DMA/THF(1/2) | trace | – |
| 13 | L15 | DMA/THF(1/2) | 78 | 91/92 |
| 14 | L16 | DMA/THF(1/2) | 63 | 84/85 |
| 15 | L17 | DMA/THF(1/2) | 43 | 8/8 |
| 16 | L18 | DMA/THF(1/2) | trace | – |
| 17[f] | L15 | DMA/THF(1/2) | 21 | 91/92 |
| 18[g] | L15 | DMA/THF(1/2) | trace | – |
| 19 | - | DMA/THF(1/2) | trace | – |

[a]Reaction conditions: *E*-**4a** (0.1 mmol, 99% ee), **5a** (0.2 mmol), Ni(ClO$_4$)$_2$·6H$_2$O (10 mol %), ligand (20 mol %), Mn dust (0.3 mmol), NaI (0.05 mmol) in a sealed tube in 2 mL solvent at 20 °C, unless noted otherwise.
[b]Isolated yields after flash chromatography.
[c]The ee values were determined by HPLC analysis with a chiral column.
[d]The reaction was performed at 0 °C.
[e]The reaction was performed at 40 °C.
[f]Without NaI.
[g]Without Mn$^0$.

absolute configuration of **7aa** was unambiguously determined by X-ray crystallographic analysis, and those of all other *N*-heterocyclic products were assigned accordingly. A limitation of our method is the use of secondary and tertiary alkyl bromides as coupling partners, which failed to provide the desired product under the standard reaction conditions.

The substrate scope with respect to 1,6-enynes was then investigated. 1,6-Enynes bearing either electron-deficient (**7ba**) or electron-rich (**7cf**) arenes at the terminus of the alkyne moiety could be converted to the desired products in good yields with remarkably high enantiospecificity. Of particular interest was the ability to accommodate aryl chloride (**7da**); no alternative competing by-products

were observed. The reaction is not restricted to the aryl groups at the terminus of the alkyne moiety. Alkyl-substituted 1,6-enynes, including 2-phenethyl, methyl, and functionalized CH$_2$OBn, were also viable substrates, leading to the corresponding products (**7el, 7fa**, and **7gf**) in good yields and high enantiospecificity. Terminal alkyne was also tolerated to give the desired product **7 ha** in high enantiospecificity and excellent stereoselectivity (*E/Z* > 20:1). Notably, the presence of a sterically hindered quaternary carbon on the ortho position of the alkyne does not impede the cyclization reaction (**7ia**). Unfortunately, only low enantiospecificity was achieved for the synthesis of the six-membered ring product **7ja**. Previously reported nickel-catalyzed stereospecific allyl-aryl cross-coupling reactions were limited to aryl-

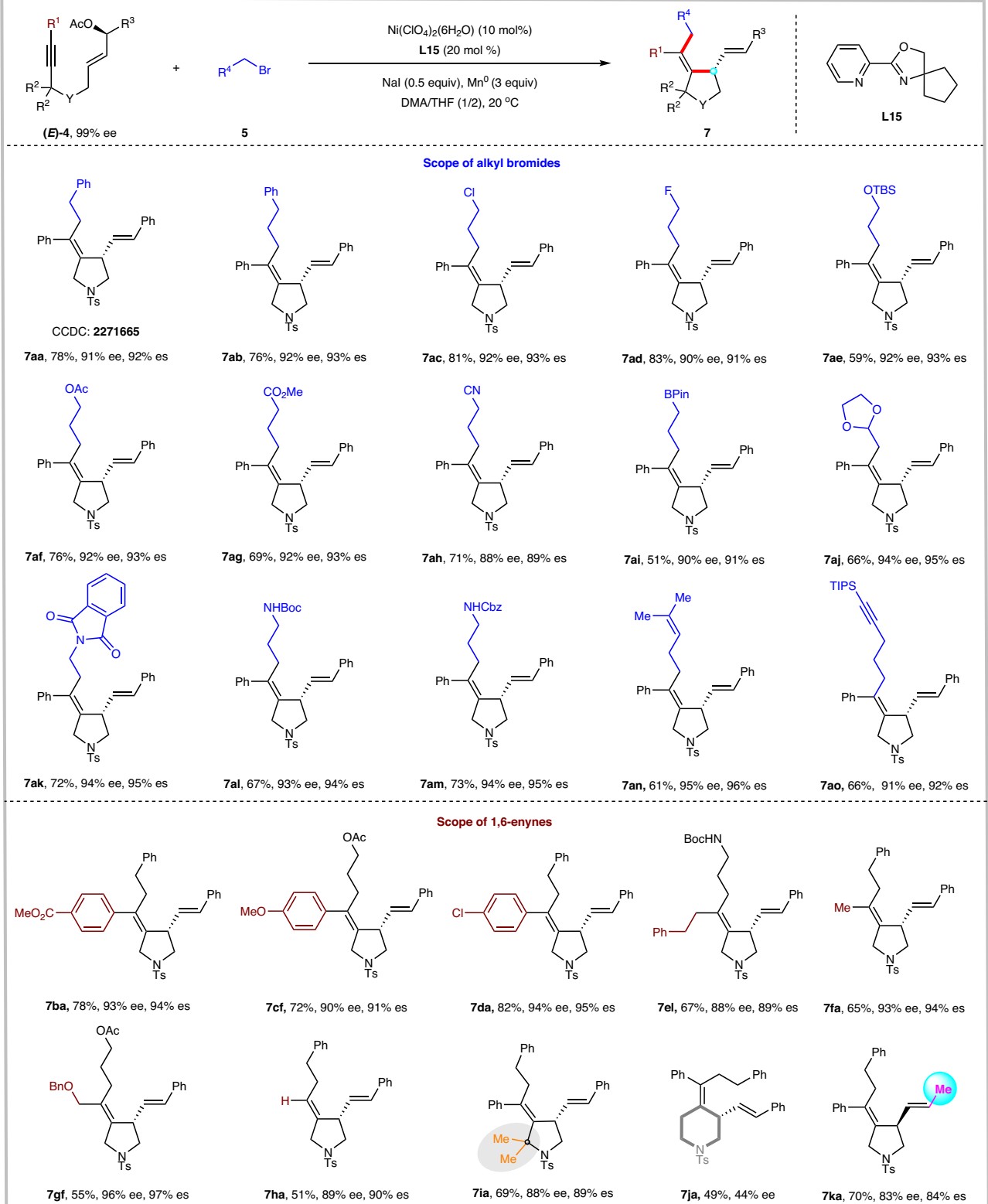

**Fig. 4 | Reaction scope of asymmetric reductive alkylative cyclization.** The reactions were performed on a 0.1 mmol scale under the conditions in Table 2, entry 13.

substituted allylic esters. Replacement of the aryl substituent (R³) on the allylic ester with an alkyl group resulted in poor regioselectivity and low enantiospecificity[44–46,73]. One possible reason is that the presence of an aryl group can stabilize the key π-allyl nickel intermediate through conjugation. Strikingly, the alkyl-substituted allylic esters

reacted smoothly to obtain the desired product **7ka** in 70% yield with slightly reduced stereospecificity (83% ee).

To explore the effect of alkene geometry on the stereochemical outcome, we compared the alkylative cyclization reactions of (*E*)-**4a** and (*Z*)-**4a** with the alkyl bromide **5a** (Fig. 5). The alkylative cyclization

**Fig. 5 | Effect of alkene geometry on stereochemical outcome.** The proposed mechanism is presented in the figure.

reactions of (*R*, *E*)-**4a** and (*S*, *E*)-**4a** with **5a** proceeded smoothly to afford **7aa** in 78–79% yields with excellent stereochemical fidelity (91–95% ee). On the other hand, the alkylative cyclization of the substrates (*R*, *Z*)-**4a** and (*S*, *Z*)-**4a** with **5a** provided **7aa** in high yields (91–94%) with slightly reduced stereochemical fidelity (79–83% ee). Clearly, the opposite absolute configuration of **7aa** was observed from the alkylative cyclization reactions of (*E*)-**4a** vs (*Z*)-**4a**, highlighting the influence of alkene geometry on the stereochemical outcome.

Based on these results, we propose that (*Z*)-**4a** and (*E*)-**4a** undergo stereo inverted oxidative addition to nickel to form π-allylnickel complexes **I** and **II**, respectively. The π-allylnickel complexes **II** undergo the isomerization of the substituent from *anti* to *syn* by way of σ-allylnickel intermediates. The π–σ–π rearrangement moves the nickel from the back side to the front side to give the π-allylnickel complex **III**, which has the same configuration as π-allylnickel complexes **I**. This is the reason why (*R*, *E*)-**4a** and (*S*, *Z*)-**4a** react with **5a** to produce the same product (*S*)-**7aa**. This phenomenon is similar to the previously reported Pd-catalyzed stereoselective nucleophilic substitution reaction of optically active allylic acetates[75,76].

## Synthetic applications

Our approach was also successfully applied to reactions using structurally complex alkyl bromides derived from estrone (**7ap**) and glucose (**7aq**) as precursors, while leaving other sensitive functional groups (ester, acetal, and ketone) intact. Our method provides a means to incorporate enantioenriched tetrahydropyrrole into biologically active molecules. For example, the alkylative cyclization of (*E*)-**4a** with alkyl bromides derived from galactose and naproxen afforded the products **7ar** and **7as** in moderate yields with high diastereoselectivities, respectively (Fig. 6a).

Moreover, the alkylative cyclization reaction of (*E*)-**4a** with (bromomethyl)cyclopropane **5t** followed by ring-closing metathesis afforded the cyclohepta[c]pyrrole skeleton **8**, which is widely found in many natural products and pharmaceuticals[77,78]. To our delight, using palladium on carbon as a catalyst for hydrogenation at one atmosphere, the less substituted double bond was selectively hydrogenated to give tetrahydropyrrole **9aa** in 91% yield while maintaining high enantioselectivity. The tetrasubstituted alkenes in **9aa** could be further hydrogenated at 10 atmospheres to give **10aa** with three chiral stereocenters with high efficiency (93% yield) and excellent asymmetric induction (>20/1 d.r.) (Fig. 6b).

To further demonstrate the utility of this protocol in the field of medicinal chemistry, the application of this approach to the concise synthesis of pharmaceutically relevant molecules was performed. Brivaracetam, released in 2016 under the brand name Briviact, is used to treat partial-onset seizures[2–5]. The alkylative cyclization of (*E*)-**4l** with **5a** afforded the 2-pyrrolidone skeleton **7la** in 65% with 82% ee. Selective hydrogenation of **7la** gave product **9la** in 99% yield. Ozonolysis of **9la** gave the ketone intermediate, which was then reduced with NaBH₄ at −78 °C to afford alcohol **11** in 72% yield[17]. **11** underwent bromination, and subsequent radical reductive debromination afforded 2-pyrrolidone **12**[79], which can be readily converted to the antiepileptic drug Brivaracetam by known procedures (Fig. 6c)[80,81].

## Mechanistic studies

To shed light on the different reaction patterns of 1,6-enynes with aryl halides and alkyl halides, we designed a series of mechanistic experiments (Fig. 7). The reaction of 1,7-enyne **1o** with **5a** successfully afforded the tetrahydropyrrole **6aa** in 70% yield, indicating that the alkylative cyclization reaction was initiated by the oxidative addition of allylic acetates to nickel to form π-allylic nickel intermediates (Fig. 7a, left). However, the reaction of **1o** with **2a** did not produce the tetrahydropyridine **3aa** or tetrahydropyrrole **3aa′**, implying that a π-allylic nickel intermediate may not be involved in the arylative cyclization process (Fig. 7a, right).

Furthermore, we synthesized alkenyl iodides **13** and **14** and subjected them to our standard reaction conditions of arylative cyclization. To our surprise, the expected product **3aa** was isolated in 49% and 58% yields, respectively, while the ee value of **3aa** was drastically reduced to 3% (Fig. 7b). If the reductive cyclization proceeds via oxidative addition of alkenyl iodide of **13** or **14** to nickel to generate alkenyl nickel, and then addition to allylic acetate followed by β-OAc elimination, the resulting product **3aa** should be highly stereoselective. Therefore, we believe that the reaction proceeds through the oxidative addition of nickel to allylic acetate to obtain allylic nickel, followed by cross-coupling with alkenyl iodide. **L8** may not be able to control the stereoselectivity of this process.

We, therefore, further designed a crossover experiment in which **1c** and **13** were reacted with PhI under standard reaction conditions (Fig. 7c). The expected product **3ca** was obtained with high enantioselectivity (94% ee); while **3aa** is almost racemic. This result further

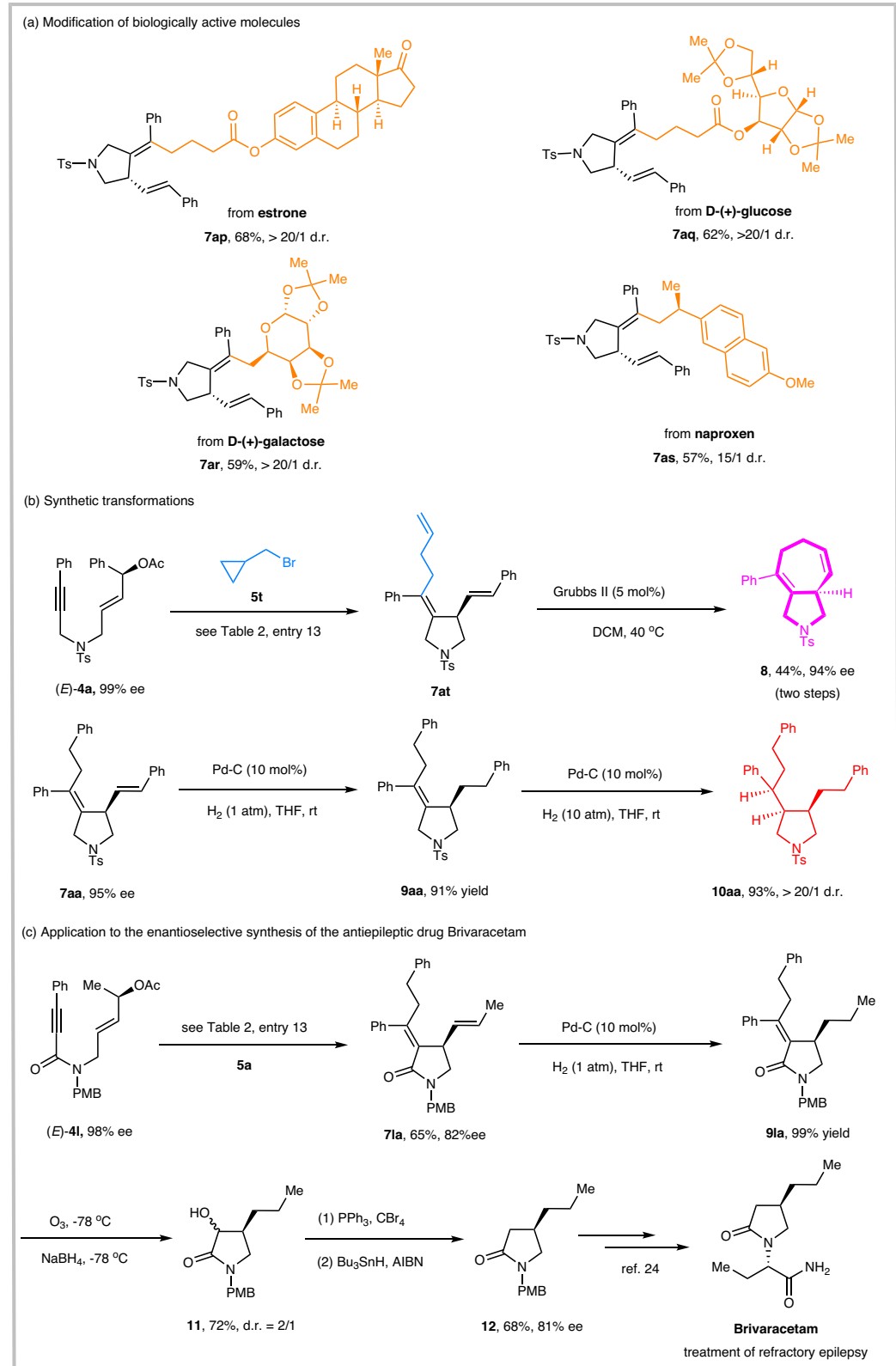

**Fig. 6 | Synthetic transformations and applications. a** Modification of biologically active molecules. **b** Synthetic transformations. **c** Application to the enantioselective synthesis of the antiepileptic drug Brivaracetam.

supports that the arylative cyclization mechanism is unlikely to proceed by first forming an allylic nickel intermediate.

To clarify possible catalytically active intermediates in the arylative cyclization reaction, we synthesized aryl-Ni(II) complex **15**[82,83] and

reacted it with **1a** to give **3ae** in 80% yield (Fig. 7d). This result suggests that aryl-nickel species is a key intermediate in the catalytic cycle.

We assumed that the alkylative cyclization occurs through the oxidative addition of nickel to allylic acetate to generate a π-allylic

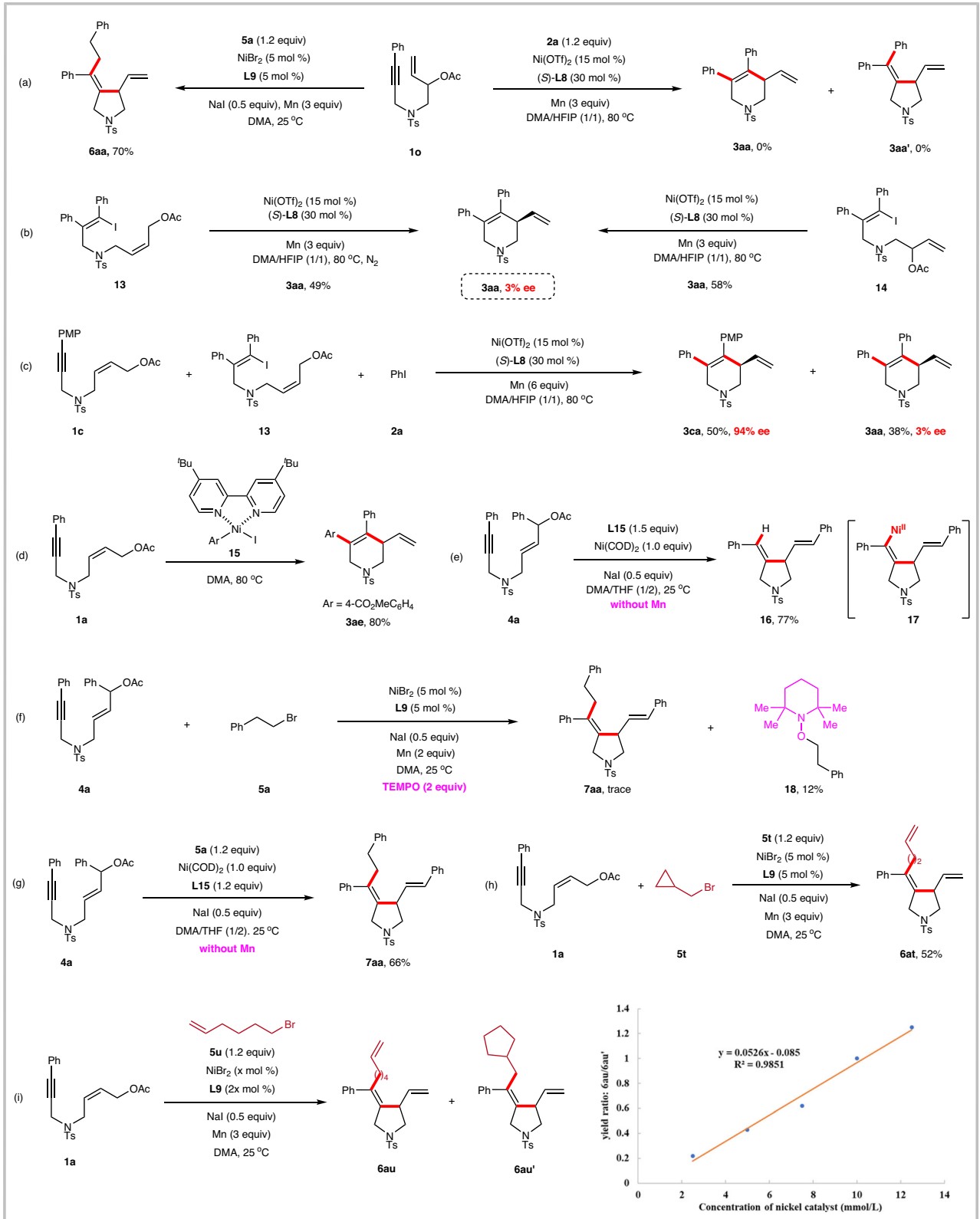

**Fig. 7 | Mechanistic investigations. a** Reductive cyclization of 1,7-enynes. **b** Reductive cyclization of alkenyl iodides. **c** Crossover experiment. **d** Stoichiometric experiment with aryl-Ni(II) complexes. **e** Quench of alkenylnickel intermediates. **f** Radical trapping experiment. TEMPO tetramethylpiperidine oxide. **g** Stoichiometric experiment with Ni(0). **h** Ring opening reaction. **i** Ring, closing reaction.

**Fig. 8 | Proposed Mechanism. a** Proposed carbometallation pathway for the enantioselective reductive *trans*-arylative cyclization. **b** Proposed allylic oxidative addition pathway for the enantiospecific reductive *cis*-alkylative cyclization.

nickel, followed by migratory insertion into the alkyne to generate the alkenyl-nickel(II) species **17**. To verify this process, we performed a stoichiometric reaction of Ni(COD)₂ with **4a** and quenched the reaction with water. The product **16** was obtained in 77% yield (Fig. 7e). This result confirms our hypothesis that alkenyl nickel(II) species **17** are indeed involved in the catalytic cycle.

When the radical scavenger TEMPO was added to standard alkylative conditions, the reaction was completely inhabited, and the TEMPO-trapped phenylethane product **18** was isolated in 12% yield (Fig. 7f), indicating the involvement of the radical pathway.

To distinguish the sequential reduction and radical chain pathways in our reductive alkylative cyclization, the reaction of **4a** with **5a** using stoichiometric Ni(0) catalyst in the absence of the reductant (Mn) was carried out, leading to the desired product **7aa** in 66% yield (Fig. 7g). This result indicates that a radical chain mechanism is reasonably proposed to be responsible for the catalytical cycle. The sequential reduction pathway involving the reduction of alkenyl nickel(II) **17** to nickel(I), oxidative addition to alkyl bromide to form Ni(III), and reductive elimination can be ruled out at this stage.

Next, we performed a series of radical clock experiments. The alkylative cyclization reaction of **1a** and cyclopropylmethyl bromide (**5t**) gave the ring-opening product **6at** in 52% yield (Fig. 7h). This result clearly shows that alkyl radical was generated from the alkyl bromide during the reaction process. Interestingly, the reaction of **1a** with 6-bromo-1-hexene (**5u**) afforded the ring-closure product **6au'** as well as the uncyclized product **6au**. We further investigated the relationship between the concentration of nickel catalyst and the product ratio of **6au** and **6au'** (Fig. 7i). A linear relationship was observed, supporting a radical chain mechanism instead of cage-bound oxidative addition.

Based on the above experimental results, possible mechanisms for the reductive arylative and alkylative cyclizations were proposed, respectively (Fig. 8). For nickel-catalyzed enantioselective reductive arylative cyclization, the oxidative addition of aryl iodides to catalytically active Ni(0) catalyst **F** would afford aryl-Ni(II) species **A**, which undergoes intermolecular migratory insertion into alkyne to provide alkenyl-Ni(II) intermediate **B**. *E/Z* isomerization of alkenyl-nickel

species **B**, involving a previously proposed possible zwitterionic carbene intermediate[19,84,85], would afford a new alkenyl-nickel intermediate **C**, which undergoes cyclization with allylic acetate to form alkylnickel intermediate **D**. Subsequent β-OAc elimination of nickel species **D** would furnish the chiral six-membered ring product **3** and Ni(II)X complex **E**, which can regenerate the active Ni(0) catalyst **F** upon Mn reduction. Arylative cyclization is initiated by the oxidative addition of nickel to aryl iodides rather than to allylic acetates, probably because aryl iodides are more reactive than allylic acetates.

For nickel-catalyzed enantiospecific reductive alkylative cyclization, stereoinvertive oxidative addition of allylic acetate to Ni(0) catalyst would afford π-allylic nickel intermediate **G**, which undergoes intramolecular stereoretentive migratory insertion into alkyne to provide alkenyl-Ni(II) intermediate **H**. Meanwhile, the in situ generated Ni(I) species **J** chemically engages a single-electron transfer event with alkyl halide **5** to give rise to alkyl radical **K** and Ni(II) complex. The radical addition of the alkyl radical **K** with the alkenyl-Ni(II) intermediate **H** would generate Ni(III) intermediate **I**. The possibility of reduction of alkenyl-Ni(II) **H** to alkenyl nickel(I) **L** followed by one-electron transfer with alkyl bromide and recombination to form Ni(III) **I** was ruled out. Subsequent reductive elimination from the Ni(III) species **I** would furnish the enantioenriched five-membered ring product **7** along with the Ni(I) intermediate **J**.

## Discussion

In summary, Ni-catalyzed enantioselective reductive *anti*-arylative cyclization of 1,6-enynes with aryl iodides was reported, providing enantioenriched six-membered carbo- and heterocycles in good yields with excellent enantioselectivities. Moreover, the enantiospecific reductive *cis*-alkylative cyclization of 1,6-enynes with alkyl bromides was also realized, furnishing chiral five-membered nitrogen-containing heterocycles in good yields with high regioselectivity and stereochemical fidelity. Mechanistic studies reveal that the arylative cyclization of 1,6-enynes is initiated by the oxidative addition of aryl halide to Ni(0), followed by carbonickelation of the alkyne, *E/Z* isomerization and enantioselective addition to allylic acetates. The alkylative

cyclization is triggered by the oxidative addition of allylic acetate to Ni(0), followed by stereospecific insertion of alkynes and coupling with alkyl halides. The utility of this strategy was further demonstrated in the enantioselective synthesis of the antiepileptic drug Brivaracetam. Further investigation on enantioselective alkylative cyclization of 1,6-enynes is still ongoing in our laboratory.

## Methods

### General procedure for the enantioselective reductive arylative cyclization of 1,6-enynes

An oven-dried sealed tube equipped with a PTFE-coated stir bar was charged with Ni(OTf)$_2$ (0.03 mmol, 10.6 mg), (S)-**L8** (0.06 mmol, 21.2 mg) and anhydrous DMA/HFIP (1:1, 2 mL). This reaction mixture was stirred at room temperature for 1 h in an argon-filled glovebox. Mn (0.6 mmol, 33.0 mg), 1,6-enyne **1** (0.2 mmol), aryl iodide **2** (0.4 mmol), and DMA/HFIP (1:1, 2 mL) were then added. The sealed tube was sealed and removed from the glovebox. The reaction mixture was allowed to stir at 80 °C for 12 h. The reaction was quenched by the addition of H$_2$O (10 mL) and EtOAc (20 mL). The organic layer was separated, and the aqueous layer was extracted with EtOAc (20 mL × 3). The combined organic layers were washed with brine, dried over Na$_2$SO$_4$, filtered, and concentrated. The residue was purified by flash chromatography on silica gel, eluting with PE/EtOAc (50/1 - 5/1), to give the desired products **3**.

### General procedure for the enantiospecific reductive alkylative cyclization of 1,6-enynes

An oven-dried sealed tube equipped with a PTFE-coated stir bar was charged with Ni(ClO$_4$)$_2$·6H$_2$O (0.01 mmol, 3.7 mg), **L15** (0.02 mmol, 4.0 mg), NaI (0.05 mmol, 7.5 mg) and anhydrous DMA (0.67 mL). This reaction mixture was stirred at room temperature for 1 hour in an argon-filled glovebox. 1,6-Enyne **4** (0.1 mmol), alkyl bromide **5** (0.2 mmol), Mn powder (0.3 mmol, 16.5 mg), and anhydrous THF (1.33 mL) were then added. The sealed tube was sealed and removed from the glovebox. The reaction mixture was allowed to stir at 20 °C until the reaction was completed (monitored by TLC). The reaction was quenched by the addition of a saturated aqueous solution of NH$_4$Cl (10 mL) and EtOAc (20 mL). The organic layer was separated, and the aqueous layer was extracted with EtOAc (20 mL × 3). The combined organic layers were washed with brine, dried over Na$_2$SO$_4$, filtered, and concentrated. The residue was purified by flash chromatography on silica gel, eluting with PE/EtOAc (20/1–5/1), to give the desired products **7**.

## Data availability

The authors declare that all the data supporting the findings of this work are available within the article and its Supplementary Information files or from the corresponding author upon request. The X-ray crystallographic coordinates for structures reported in this study have been deposited at the Cambridge Crystallographic Data Center (CCDC) under deposition numbers 2283739 (**3aj**) and 2271665 (**7aa**). These data can be obtained free of charge from The Cambridge Crystallographic Data Center via www.ccdc.cam.ac.uk/data_request/cif.

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

## Acknowledgements

We are grateful for financial support from the National Natural Science Foundation of China (21901192 (K.S.) and 22271225 (K.S.)), the Natural Science Foundation of Hubei Province (2019CFB129 (K.S.) and 2022CFB221 (K.S.)), and the Hubei Provincial Outstanding Youth Fund (2022CFA092 (W.K.)).

## Author contributions

W.L., Y.X., and D.Y. contributed equally. W.K. and K.S. conceived and designed the project. W.L., Y.X., and D.Y. conducted the experiments. W.K., K.S., and W.L. analyzed and interpreted the experimental data. W.K. and K.S. prepared the paper. W.L., Y.X., and D.Y. prepared the Supplementary Information. All authors contributed to discussions.

## Competing interests

The authors declare no competing interests.
