## [Peer Review File · Nature Communications]

Nickel-Catalyzed Electrophiles-Controlled Enantioselective Reductive Arylative Cyclization and Enantiospecific Reductive Alkylative Cyclization of 1,6-EnynesREVIEWER COMMENTS

Reviewer #1 (Remarks to the Author):

The present study highlights the reductive coupling of 1,6-enynes featuring allylic acetate moieties with alkyl/aryl halides under the Ni-catalyzed cross-electrophile frameworks. The use of chiral Ni-catalyzed assembled by Ni and a P,N-ligand enabled excellent stereocontrol of the tetrahydropyridine products. The same protocol was not applicable for alkyl halides, but the authors were able to attain enantiospecific reductive cis-alkylative cyclization of 1,6-enynes bearing chiral allyl acetates with primary alkyl bromides to afford chiral tetrahydropyrroles with high regioselectivity. The cyclization of 1,6-enynes decorated with allylic acetates/phosphates have been relatively well-explored for their coupling with arylboronic acids under Ni-catalysis. However, the reductive coupling has not been revealed. The scope of the substrates and the types of functional groups survived in the present method are impressive as demonstrated by the large set of examples shown in the paper. I suggest publication of this work. But I do have some concerns on the proposed mechanisms.

1. The exclusion of vinyl-Ni species in the cyclization process seems to be arbitrary. First, conversion of 14 to 3aa does not necessarily involve 15. In other words, the control reaction may proceed first via allyl-Ni intermediate. Second, interconversion of 15 to 17 and to 18 is possible. However, evidence or a reference is required to support the ensuing migrative insertion process. I suggest the authors be cautious to exclude the mechanism of insertion of vinyl-Ni to allyl double bond in Scheme 8.

2. Scheme 7c, please check whether ArI was formed due to retro-reductive elimination of the Ar-Ni(II)X complex at 80 °C. If so, a chance of formation of allyl-Ni prior to cyclization and coupling is possible. Is it possible to trap an intermediate of insertion of Ar-Ni to alkyne at room temperature with the allylic moiety being intact after aqueous work-up?

3. In Scheme 5, complexes II and III that are derived from (R,E)-4a and (S,Z)-4a are suggested to undergo convergent cyclization/coupling to give (S)-7aa as the primary isomer. It is however, hard to correlate this speculation to the proposed reaction mechanism shown in Scheme 8. Based on the authors' mechanistic scenario, both II and III should have the same opportunities or overcome the same energy barrier to cyclize, and afford the complex H. Thus, the two opposite enantiomers form in the end. Perhaps for the alkylative cyclization/coupling process, an initial radical addition to the alkyne moieties (or associated with an alkyl-Ni intermediate) to give alkenyl radical that combines with Ni to afford alkenyl-Ni followed by a migrative insertion to the double bond of the allyl portion should be considered.

4. In the following sentence, "Since the pioneering work of the groups of Reisman and Weix, this strategy has emerged as an efficient and practical method for the enantioselective coupling of alkyl electrophiles." The authors may have noticed that Weix are not the first to work C(sp³)-C(sp³) XEC, and Reisman's work was significantly inspired by Greg Fu.

Reviewer #2 (Remarks to the Author):

In the present manuscript, the authors developed a methodology to construct various six-membered carbo and heterocycles in an enantioselective way and five-membered carbo and heterocycles in an enantiospecific way. An extensive substrate scope for both the arylative and alkylative reductive coupling pathways was shown. Several late-stage functionalizations and antiepileptic drug Brivaracetam synthesis have been performed. Previously, Pd, Rh, and

Ni catalyzed coupling of asymmetric cis-arylate cyclization of 1,6-enynes with aryl-boronic acids were reported to form five-membered heterocycles. In this manuscript, the authors claim to have overcome all challenges for coupling aryl and alkyl halide. However, this comes with the cost of preparation of an alkyne-tethered allyl acetate, which might decrease the process's economy. Besides, the reaction also required 3 equivalents of Mn as reductant, producing the same metal salt equivalent as the side product. Furthermore, the asymmetric version of the reaction required 15 mol% Ni and 30 mol% of chiral ligand loading. The yields are in the range of 70-80%. That corresponds to a turnover of ~5. Thus, in terms of synthetic advancement, the process is unsustainable. Can this be improved? Besides, the cyclization of 1,6-enynes was recently coupled with C-H bond functionalization in a highly enantioselective manner (eg: J. Am. Chem. Soc., 2020, 142, 9510). Considering all these, the reviewer thus does not support the publication of this manuscript in this journal. Other comments:

- 1) Has the author tried any other leaving group instead of acetate? Did they also try other leaving groups to achieve the metal π -allyl species?
- 2) Have they tried any terminal alkyne or alkyl group?
- 3) The author used NaI as a sub-stoichiometric additive for the alkylative reductive coupling reaction. The optimization table indicates the importance of using NaI to achieve good yield. What is the actual role of this additive? Does NaI undergo a halide exchange reaction with the alkyl bromide reagent to produce alkyl iodide? Did the author try the alkylative reductive coupling reaction using alkyl iodides? Did the author try alkyl halide sources with only an alkyl chain like 1-bromobutane?
- 4) Which step is the rate-determining step? Why is such a large amount of catalyst required?
- 5) To get a clear idea about the formation of alkyl radicals, the author can do one radical trapping experiment by using an external radical scavenger.
- 6) The author should incorporate a detailed optimization table for each factor in the supplementary information. Though a ligand optimization has been displayed for alkylative cyclization, no optimization table is shown for the other case. The author should include a separate optimization table for metals, ligands, solvents, reductants, and additives.
- 7) Have the Authors isolated any intermediate or experimental evidence for Ni-carbene species and the alkylative cyclization pathway? This is crucial to establish the claim.
- 8) During the synthesis of Brivaracetam, the ee significantly drops in the first step. Any reason for that? The synthesis looks very poor atom economic, as a major part of it is being chopped out.

Reviewer #3 (Remarks to the Author):

This manuscript describes enantioselective conversion of 1,6-enynes to heterocyclic motifs using a simple nickel catalyst. Starting materials for this transformation are relatively easy to access and range of nitrogen heterocycles accessed is diverse. Overall, the manuscript is well written and organized and supporting information provide sufficient details for reproducibility of the results. This referee had would like the following questions to be addressed before publication:

1. For the tetrahydropyridines scope presented in Scheme 3 what happens if you start from alkyl (or TMS) substituted alkyne? Does the reaction work at all? If not what is the outcome? (recovered starting material, decomposition etc.)
2. In Scheme 7a right side do you observe just recovery of the starting materials or some a different product (other than 3aa and 3aa') formed.
3. For the reactions that use alkyl bromides to form pyrrolydines, what is the role NaI? As the

secondary and tertiary alkyl bromides don't work under these conditions (line 211 and 212)
is it possible that you have initial reaction of alkyl bromides with NaI to form alkyl iodides
which then participate in the reaction?

Point-by-point response to the reviewers' comments

Reviewer #1 (Remarks to the Author):

*The present study highlights the reductive coupling of 1,6-enynes featuring allylic acetate moieties with alkyl/aryl halides under the Ni-catalyzed cross-electrophile frameworks. The use of chiral Ni-catalyzed assembled by Ni and a P,N-ligand enabled excellent stereocontrol of the tetrahydropyridine products. The same protocol was not applicable for alkyl halides, but the authors were able to attain enantiospecific reductive cis-alkylative cyclization of 1,6-enynes bearing chiral allyl acetates with primary alkyl bromides to afford chiral tetrahydropyrroles with high regioselectivity. The cyclization of 1,6-enynes decorated with allylic acetates/phosphahtes have been relatively well-explored for their coupling with aryl-boric acids under Ni-catalysis. **However, the reductive coupling has not been revealed. The scope of the substrates and the types of functional groups survived in the present method are impressive as demonstrated by the large set of examples shown in the paper. I suggest publication of this work.** But I do have some concerns on the proposed mechanisms.*

We thank the reviewers for their very positive comments.

- 1. The exclusion of vinyl-Ni species in the cyclization process seems to be arbitrary. First, conversion of **14** to **3aa** does not necessarily involve **15**. In other words, the control reaction may proceed first via allyl-Ni intermediate. Second, interconversion of **15** to **17** and to **18** is possible. However, evidence or a reference is required to support the ensuing migrative insertion process. I suggest the authors be cautious to exclude the mechanism of insertion of vinyl-Ni to allyl double bond in Scheme 8.*
- 2. Scheme 7c, please check whether ArI was formed due to retro-reductive elimination of the Ar-Ni(II)X complex at 80 °C. If so, a chance of formation of allyl-Ni prior to cyclization and coupling is possible. Is it possible to trap an intermediate of insertion of Ar-Ni to alkyne at room temperature with the allylic moiety being intact after aqueous work-up?*

Answer: These are two very good questions. Considering that your main concern is that the arylation cyclization is initiated by the oxidative addition of nickel to allyl acetates, we decided to combine the answers to these two questions.

(1) We synthesized alkenyl iodide **14** and found that it can indeed undergo cyclization reaction to obtain the expected product **3aa**. However, the ee value of **3aa** was drastically reduced to 3%.

If the reductive cyclization proceeds via oxidative addition of alkenyl iodide of **13** or **14** to nickel to generate alkenyl nickel, and then addition to allylic acetate followed by β -OAc elimination, the resulting product **3aa** should be highly stereoselective.

Therefore, we believe that the reaction proceeds through the oxidative addition of nickel to allylic acetate to obtain allylic nickel, followed by cross-coupling with alkenyl iodide. **L8 cannot control the stereoselectivity of this process.**

We therefore further designed a crossover experiment in which **1c** and **13** were reacted with PhI under standard reaction conditions (Scheme 7c). The expected product **3ca** was obtained with high enantioselectivity (94% ee); while **3aa** is almost racemic. **This result clearly shows that the two products 3ca and 3aa are obtained through different mechanisms.**

high enantioselective process

Based on this result, we believe that our previously proposed cyclization mechanism based on nickel carbene species is too hasty. Therefore, we removed the representation of the nickel carbene intermediate. We thank the reviewers for their very constructive suggestions.

(2) The reaction of **1o** with PhI did not produce the tetrahydropyridine **3aa** or tetrahydropyrrole **3aa'**, implying that a π -allylic nickel intermediate may not be involved in the arylation cyclization process. If the arylation cyclization proceeds first via an allyl-Ni intermediate, it should be possible to obtain products **3aa** or **3aa'**.

The reaction of 1,7-enyne **1o** with alkyl bromide **5a** successfully afforded the tetrahydropyrrole **6aa** in 70% yield, indicating that the alkylative cyclization reaction was initiated by the oxidative addition of allylic acetates to nickel to form π -allylic nickel intermediates.

Putting the two mechanisms together may be more intuitive. If the arylation cyclization occurs first through the oxidative addition of nickel to allyl acetates to form allylic nickel, there are two cyclization modes. It is obvious that *exo-dig*-cyclization is more favorable. *Endo-dig*-cyclization is difficult to proceed with high selectivity.

(3) To trap the alkenyl-Ni intermediate generated by insertion of Ar-Ni to alkyne, the reaction of Ar-Ni(II)X complex with **1a** was conducted at room temperature for 2 hours followed by aqueous work-up. As shown below, the cyclization product **3ae** was isolated in 33% yield, but the protonated product was not detected.

(4) Nickel-catalyzed intramolecular allylic alkenylation enabled by reversible alkenylnickel *E/Z* isomerization has been well-established by Lam and other groups, representative references are as follows: *Angew. Chem. Int. Ed.* **2017**, *56*, 8216; *J. Am. Chem. Soc.* **2016**, *138*, 8068; *Chem. Sci.* **2016**, *7*, 5815; *Angew. Chem. Int. Ed.* **2019**, *58*, 15808; *Chem. Sci.* **2020**, *11*, 10204.

Taking together, the oxidative addition of nickel to aryl iodides and allyl acetates is both feasible. We believe that arylytic cyclization is initiated by the oxidative addition of nickel to aryl iodides rather than to allylic acetates, probably because aryl iodides are more reactive than allylic acetates.

3. In Scheme 5, complexes II and III that are derived from (*R,E*)-**4a** and (*S,Z*)-**4a** are suggested to undergo convergent cyclization/coupling to give (*S*)-**7aa** as the primary isomer. It is however, hard to correlate this speculation to the proposed reaction mechanism shown in Scheme 8. Based on the authors' mechanistic scenario, both II and III should have the same opportunities or overcome the same energy barrier to cyclize, and afford the complex H. Thus, the two opposite enantiomers form in the end. Perhaps for the alkylative cyclization/coupling process, an initial radical addition to the alkyne moieties (or associated with an alkyl-Ni intermediate) to give alkenyl radical that combines with Ni to afford alkenyl-Ni followed by a migrative insertion to the double bond of the allyl portion should be considered.

Answer:

(1) It is important to understand that, the configuration of initially generated 1,3-substituted π -allylic nickel II is designated as "syn, anti", which is thermodynamically unstable and can undergoes syn/anti exchange via π - σ - π rearrangement, resulting (syn, syn)-III with inversion of configuration.

It should be noted that the π - σ - π rearrangement of II is faster than its direct cyclization process, thus (*R,E*)-**4a** and (*S,Z*)-**4a** would form the same enantiomer.

This phenomenon is similar to the previous reported Pd-catalyzed stereoselective allylic substitution reaction. Hayashi and Trost provided a more detailed explanation, see the references below (Ref 75 and 76 in the main text): Hayashi, T. et al. *J. Org. Chem.* **1986**, *51*, 723-727; Trost, B. M. et al. *Chem. Rev.* **1996**, *96*, 395-422.

(2) The suggested alkyl radical addition/cyclization pathway is unlikely. **Intermolecular radical addition of internal alkynes is difficult and often exhibits very poor regioselectivity for asymmetric dialkyl alkynes. This contrasts with our observation that substrates such as 7eI, 7fa, and 7gf exhibit unique regioselectivity.**

Furthermore, mechanistic experiments (Scheme 7f) clearly show that the alkylation reaction is initiated by allyl acetate. In the absence of alkyl halide, we can obtain the corresponding hydrolysis product **20**.

Last but not least, if the mechanism is alkyl radical addition followed by cyclization, then it is impossible to obtain **6aa** by reacting **1o** with **5a** under standard conditions (Scheme 7a).

4. In the following sentence, “Since the pioneering work of the groups of Reisman and Weix, this strategy has emerged as an efficient and practical method for the enantioselective coupling of alkyl electrophiles.” The authors may have noticed that Weix are not the first to work C(sp³)-C(sp³) XEC, and Reisman’s work was significantly inspired by Greg Fu.

Answer: Thank you for your friendly reminder. Seminal reports by Semmelhack (*J. Am. Chem. Soc.* **1971**, *93*, 5908), Kende (*Tetrahedron Lett.* **1975**, *16*, 3375), and Kumada (*Tetrahedron Lett.* **1977**, *18*, 4089) demonstrated the ability of nickel to mediate the reductive homocoupling of C(sp²) halide electrophiles to form biaryl products. In 2007, Durandetti and co-workers reported the Ni-catalyzed reductive C(sp²)-C(sp³) cross-coupling of α -chloro esters and aryl iodides using Mn⁰ as a terminal reductant. Weix, Gong, and other groups followed in 2010 with the Ni-catalyzed reductive cross-coupling a sec-alkyl bromide and an aryl iodide, also utilizing a Ni(II) catalyst and bipyridine-based ligand. Indeed, Reisman’s work was inspired by Greg Fu on Ni-catalyzed enantioconvergent redox-neutral cross-couplings.

We therefore revised this sentence as follows:

“Over the past decade, there has been a surge of interest in Ni-catalyzed reductive cross-coupling reactions, a strategy that allows reactions to be performed under mild conditions with high functional group tolerance and avoids the handling of sensitive organometallic reagents (Scheme 2b).^{47–57} This strategy has emerged as an efficient and practical method for the enantioselective coupling of alkyl electrophiles.^{58–62}”

Reviewer #2 (Remarks to the Author):

In the present manuscript, the authors developed a methodology to construct various six-membered carbo and heterocycles in an enantioselective way and five-membered carbo and heterocycles in an enantiospecific way. An extensive substrate scope for both the arylative and alkylative reductive coupling pathways was shown. Several late-stage functionalizations and antiepileptic drug Brivaracetam synthesis have been performed. Previously, Pd, Rh, and Ni catalyzed coupling of asymmetric cis-arylate cyclization of 1,6-enynes with aryl-boronic acids were reported to form five-membered heterocycles. In this manuscript, the authors claim to have overcome all challenges for coupling aryl and alkyl halide. However, this comes with the cost of preparation of an alkyne-tethered allyl acetate, which might decrease the process's economy.

Answer: Alkyne-tethered allylic alcohol derivatives, such as allylic ethers, acetates, carbonates and phosphates, are easily to be prepared, and they have been employed in previously reported Pd, Rh, and Ni-catalyzed arylative cyclization of 1,6-enynes. Enantioenriched secondary allylic alcohols are also readily available, and enantiospecific allylic substitution has been explored by other groups (see ref. 31–46).

Moreover, enantiospecific allylic substitution takes the advantage of the avoidance of chiral ligands. To improve the step economy of our method, we have tested the alkylative cyclization using alkyne-tethered allylic alcohol as substrate, which was transformed into the corresponding allylic acetate in situ in the presence of Ac_2O (see below). However, a lot of the protonated by-product was produced, and the yield of the desired product was very low.

Besides, the reaction also required 3 equivalents of Mn as reductant, producing the same metal salt equivalent as the side product.

Answer: Previously reported arylative cyclization of 1,6-enynes typically use organometallic reagents to initiate the reaction. Organometallic reagents require additional steps to be pre-prepared from the corresponding halides, and their use significantly reduces the functional group tolerance of the reaction.

Although our reductive cyclizations require metal reducing agents, these reactions avoid the preparation and handling of air- and moisture-sensitive organometallic reagents. And compared

with organometallic reagents, organic electrophiles such as aryl halides and alkyl halides are relatively more stable, easy to handle, and easy to obtain.

Furthermore, the asymmetric version of the reaction required 15 mol% Ni and 30 mol% of chiral ligand loading. The yields are in the range of 70-80%. That corresponds to a turnover of ~5. Thus, in terms of synthetic advancement, the process is unsustainable. Can this be improved?

Answer: The racemization reaction only requires 5 mol% nickel catalyst to achieve a yield of 71%. Reactions using chiral ligands often result in reduced activity, possibly due to steric hindrance of the chiral ligands. We used 15 mol% Ni and 30 mol% chiral ligand mainly to balance the contradiction between yield and stereoselectivity.

Besides, the cyclization of 1,6-enynes was recently coupled with C-H bond functionalization in a highly enantioselective manner (eg: *J. Am. Chem. Soc.* 2020, 142, 9510). Considering all these, the reviewer thus does not support the publication of this manuscript in this journal.

Answer: Regarding the elegant work reported by Lautens on Co-catalyzed enantioselective hydroarylation of 1,6-enynes **involving a directing group-assisted** C-H bond activation (*J. Am. Chem. Soc.* 2020, 142, 9510), **this still falls into the category of arylative cyclization of 1,6-enynes**, where the aryl group is introduced not through transmetalation with boronic acid but through **pyridine-directed** C-H bond activation.

Transition metal-catalyzed asymmetric arylation cyclization of 1,6-enynes with aryl-boronic acids to form five-membered heterocycles has been well studied. Despite remarkable achievements have been made in this field, there are still some considerable limitations.

(a) These transformations are essentially restricted to the use of aryl or alkenyl metal reagents with limited functional group compatibility. (b) Stereoselective or stereospecific alkylation cyclization of 1,6-enynes remains unexploited, probably due to the easy β -H elimination of the alkyl metal intermediate.

We believe that our work has made significant breakthroughs in the above two aspects:

- (1) We reported Ni-catalyzed enantioselective reductive *anti*-arylation cyclization of 1,6-enynes with aryl iodides, providing enantioenriched six-membered carbo- and heterocycles in good yields with excellent enantioselectivities. This transformation shows high functional group tolerance.
- (2) **Stereoselective or stereospecific alkylation cyclization of 1,6-enynes has not yet been achieved.** We therefore realized **the first example of** Ni-catalyzed enantiospecific reductive *cis*-alkylation cyclization of 1,6-enynes with alkyl bromides, furnishing chiral five-membered heterocycles with high regioselectivity and stereochemical fidelity. The alkylation cyclization is triggered by the oxidative addition of Ni(0) to allylic acetates to generate π -allylic intermediates, followed by stereospecific insertion of alkynes and coupling with alkyl halides. The utility of this strategy was further demonstrated in the enantioselective synthesis of the antiepileptic drug Brivaracetam.

Other comments:

(1) *Has the author tried any other leaving group instead of acetate? Did they also try other leaving groups to achieve the metal π -allyl species?*

Answer: In addition to allylic acetates, we also examined other leaving groups in arylation and alkylation catalytic systems, including allylic phosphonate and allylic carbonate. These results are as follows. In comparison, allylic acetate works best, which is why we use allyl acetate. These results were added into the revised Supporting Information (see S44 for more details).

(2) Have they tried any terminal alkyne or alkyl group?

Answer: For nickel-catalyzed reductive alkylative cyclization, terminal alkyne or alkyl-substituted enynes are compatible, see products **7el**, **7fa**, **7gh** and **7ha** in Scheme 4.

For nickel-catalyzed reductive arylyative cyclization, when terminal alkynes are used, most of the starting material was decomposed and only 19% of the reductive cyclization product is obtained. Alkyl-substituted alkyne give a 1:1 mixture of regioisomers.

The influence of substrate structure on the reaction further indicates that reductive arylyative cyclization and reductive alkylative cyclization may undergo two different reaction mechanisms.

(3) The author used NaI as a sub-stoichiometric additive for the alkylative reductive coupling reaction. The optimization table indicates the importance of using NaI to achieve good yield. What is the actual role of this additive? Does NaI undergo a halide exchange reaction with the alkyl bromide reagent to produce alkyl iodide? Did the author try the alkylative reductive coupling reaction using alkyl iodides? Did the author try alkyl halide sources with only an alkyl chain like 1-bromobutane?

Answer:

(1) According to previous reports (*J. Org. Chem.* **1986**, *51*, 2627; *Chem. Commun.* **2010**, *46*, 5743; *J. Am. Chem. Soc.* **2014**, *136*, 14365), iodide salts (NaI, KI, TBAI...) can accelerate the electron transfer between the metal reducing agent and the nickel catalyst, so it is widely used in nickel catalytic reduction cross-coupling.

(2) To test the possibility of halide exchange between alkyl bromides and NaI, we used alkyl iodide as an alkylation source in the presence or absence of NaI. The product **7aa** could be obtained in 72% and 55% yield, respectively. **By comparing these results of alkyl bromide with alkyl iodide, we cannot rule out the possibility of halide exchange between alkyl bromides with NaI.**

(3) According to your requirements, using 1-bromobutane as the alkyl halide source, the product **7aj** can also be obtained in 75% yield and 92% ee. This result further demonstrates the broad substrate scope of this transformation. These results have been added into the revised Supporting Information (see S45 for more details).

(4) Which step is the rate-determining step? Why is such a large amount of catalyst required?

Answer: In the reductive arylyative cyclization reaction, both (*Z*)-**1a** and (*E*)-**1a** gave comparable yields in their reactions with **2a**, indicating that the stereochemistry of the allylic acetate had little effect on the reaction efficiencies. This result suggests that the intramolecular migratory alkene insertion of alkenyl-Ni(II) species, generated from the formal *trans*-carbonickelation of alkyne, into the allylic acetate moiety might not be the rate-determining step. The yields of the arylyative cyclization were largely affected by the electronic properties of the aryl substituent on the alkyne moiety, **suggesting that the intermolecular carbonickelation might be the rate determining step in the reductive arylyative cyclization reaction.**

In the reductive alkylative cyclization, the reaction was initiated by the oxidative addition of allylic acetates, followed by intramolecular cyclization and intermolecular reductive cross-coupling with alkyl halides. The oxidative addition and intramolecular cyclization steps might

not be the rate-determining steps, as the reaction efficiencies were not significantly affected by the configuration of the alkene moiety and the electronic properties of the phenyl groups on the alkyne moiety. **Considering that secondary alkyl halides were ineffective in the reductive alkylative cyclization, we speculated that the intermolecular reductive cross-coupling with alkyl halides would be the rate-determining step.**

(5) *To get a clear idea about the formation of alkyl radicals, the author can do one radical trapping experiment by using an external radical scavenger.*

Answer: According to your suggestion, the radical scavenger TEMPO was added into standard alkylative reaction conditions. The reaction was almost completely suppressed and the TEMPO-trapped phenylethane product was isolated in 12% yield, indicating that a radical pathway was involved. These results have been added into the revised Supporting Information (see S170 for more details).

(6) *The author should incorporate a detailed optimization table for each factor in the supplementary information. Though a ligand optimization has been displayed for alkylative cyclization, no optimization table is shown for the other case. The author should include a separate optimization table for metals, ligands, solvents, reductants, and additives.*

Answer: Ligands and solvents have the most pronounced effects on the stereospecificity of alkylation cyclizations, and **these results are clearly demonstrated in Table 2.**

As your request, optimization of other influencing factors such as metals, reductants, and additives has been added into the Supporting Information (see S41-S43 for more details).

(7) *Have the Authors isolated any intermediate or experimental evidence for Ni-carbene species and the alkylative cyclization pathway? This is crucial to establish the claim.*

Answer: Thanks for this suggestion. We tried to obtain a nickel intermediate but failed.

Recent DFT calculations (representative references, see: *J. Am. Chem. Soc.* **2022**, *144*, 10064-10074; *Org. Chem. Front.* **2023**, *10*, 4243-4249; *Org. Chem. Front.* **2023**, *10*, 4263-4274) support a mechanism in which alkyne coordination to the nickel center as a η^2 -type ligand assists the *Z/E* isomerization. This η^2 -vinyl-nickel type *Z/E* isomerization is found to be lower in energy than carbonyl-assisted *Z/E* isomerization from a syn-alkenylnickel intermediate, providing new insights into the mechanism of Ni-catalyzed anti-carbometallative cyclization of alkyne.

Considering the lack of conclusive evidence, we modified the reaction mechanism based on reviewer 1's question (See our responses to reviewers for details) and previously reported DFT calculation results.

(8) During the synthesis of Brivaracetam, the *ee* significantly drops in the first step. Any reason for that? The synthesis looks very poor atom economic, as a major part of it is being chopped out.

Answer: Previously reported nickel-catalyzed stereospecific allyl-aryl cross-coupling reactions were limited to aryl-substituted allylic esters. Replacement of the aryl substituent (R^3) on the allylic ester with an alkyl group resulted in poor regioselectivity and low enantiospecificity (selected

references, see: *J. Am. Chem. Soc.* **2016**, *138*, 11989; *J. Am. Chem. Soc.* **2021**, *143*, 15930). **One possible reason is that the presence of aryl group can stabilize the key π -allyl nickel intermediate through conjugation, thereby reducing racemization.**

We performed the reaction successfully using alkyl-substituted allyl esters, affording the desired product **7ka** in 70% yield with slightly reduced stereospecificity (83% ee). This is the first example of a nickel-catalyzed coupling reaction using allylic ester with an alkyl group. **Although this result is not perfect, it is already a big improvement compared to previous works!**

Although our synthetic route appears to be short on atom economy, the showcase of such a method strengthens the utility of our enantiospecific reductive alkylative cyclization protocol in the field of medicinal chemistry and may serve as an alternative approach to Brivaracetam.

Reviewer #3 (Remarks to the Author):

This manuscript describes enantioselective conversion of 1,6-enynes to heterocyclic motifs using a simple nickel catalyst. Starting materials for this transformation are relatively easy to access and range of nitrogen heterocycles accessed is diverse. **Overall, the manuscript is well written and organized and supporting information provide sufficient details for reproducibility of the results.** This referee had would like the following questions to be addressed before publication:

We thank the reviewers for their very positive comments.

1. For the tetrahydropyridines scope presented in Scheme 3 what happens if you start from alkyl (or TMS) substituted alkyne? Does the reaction work at all? If not what is the outcome? (recovered starting material, decomposition etc.)

Answer: Based on your suggestion, we performed the following two reactions. Specifically, alkyl-substituted alkyne provided a mixture of regioisomers, while TMS-substituted alkyne successfully provided the desired product in 44% yield, albeit with moderate enantioselectivity.

2. In Scheme 7a right side do you observe just recovery of the starting materials or some a different product (other than 3aa and 3aa') formed.

Answer: The starting material **1o** was completely decomposed and only a mixture of compounds of uncertain structure was isolated, but we are very sure that **3aa** and **3aa'** were not produced.

3. For the reactions that use alkyl bromides to form pyrrolydines, what is the role NaI? As the secondary and tertiary alkyl bromides don't work under these conditions (line 211 and 212) is it possible that you have initial reaction of alkyl bromides with NaI to form alkyl iodides which then participate in the reaction?

Answer: According to previous reports (*J. Org. Chem.* **1986**, *51*, 2627; *Chem. Commun.* **2010**, 46, 5743; *J. Am. Chem. Soc.* **2014**, *136*, 14365), iodide salts (NaI, KI, TBAI...) can accelerate the electron transfer between the metal reducing agent and the nickel catalyst, so it is widely used in nickel catalytic reduction cross-coupling.

To test the possibility of halide exchange between alkyl bromides and NaI, we used alkyl iodide as an alkylation source in the presence or absence of NaI. The product **7aa** could be obtained in 72% and 55% yield, respectively. **By comparing these results of alkyl bromide with alkyl iodide, we cannot rule out the possibility of initial reaction of alkyl bromides with NaI to form alkyl iodides.**

REVIEWERS' COMMENTS

Reviewer #1 (Remarks to the Author):

I appreciate the effort that authors have put to improve the quality of the manuscript, and believe the current version of this paper is suitable for publication. The isomerization of vinyl-Ni from B to C may involve a carbene intermediate as proposed previously. See the note 19 in *Angew. Chem. Int. Ed.* 2016, 55, 15544, and reference therein (Formation of the E-products from the Z-vinyl halides may involve a possible zwitterionic C(carbene)-Ni complex, see: a) J. M. Huggins, R. G. Bergman *J. Am. Chem. Soc.* 1981, 103, 3002. b) C. Clarke, C. A. Incerti-Pradillos, H. W. Lam, *J. Am. Chem. Soc.* 2016, 138, 8068).

Reviewer #2 (Remarks to the Author):

The authors took the revision process seriously and improved the quality and scholarly presentation of the manuscript. Publication in this journal is recommended.

Reviewer #3 (Remarks to the Author):

The authors have appropriately addressed the comments by this reviewer and in my opinion the manuscript is now in a good shape to be published.

Point-by-point response to the reviewers' comments

Reviewer #1 (Remarks to the Author):

I appreciate the effort that authors have put to improve the quality of the manuscript, and believe the current version of this paper is suitable for publication.

The isomerization of vinyl-Ni from B to C may involve a carbene intermediate as proposed previously. See the note 19 in Angew. Chem. Int. Ed. 2016, 55, 15544, and reference therein (Formation of the E-products from the Z-vinyl halides may involve a possible zwitterionic C(carbene)-Ni complex, see: a) J. M. Huggins, R. G. Bergman J. Am. Chem. Soc. 1981, 103, 3002. b) C. Clarke, C. A. Incerti-Pradillos, H. W. Lam, J. Am. Chem. Soc. 2016, 138, 8068).

Answer: Thank you for your suggestion. We have cited the above references (Angew. Chem. Int. Ed. 2016, 55, 15544; J. Am. Chem. Soc. 1981, 103, 3002; J. Am. Chem. Soc. 2016, 138, 8068) in the mechanism description section.

Reviewer #2 (Remarks to the Author):

The authors took the revision process seriously and improved the quality and scholarly presentation of the manuscript. Publication in this journal is recommended.

Reviewer #3 (Remarks to the Author):

The authors have appropriately addressed the comments by this reviewer and in my opinion the manuscript is now in a good shape to be published.